# Genome-Scale Analysis of *Acetobacterium woodii* Identifies Translational Regulation of Acetogenesis

Jongoh Shin,[a] Yoseb Song,[a] Seulgi Kang,[a] Sangrak Jin,[a] Jung-Kul Lee,[b] Dong Rip Kim,[c] Suhyung Cho,[a,d] Volker Müller,[e] Byung-Kwan Cho[a,d,f]

[a]Department of Biological Sciences, Korea Advanced Institute of Science and Technology, Daejeon, Republic of Korea
[b]Department of Chemical Engineering, Konkuk University, Seoul, Republic of Korea
[c]Department of Mechanical Engineering, Hanyang University, Seoul, Republic of Korea
[d]Innovative Biomaterials Research Centre, KI for the BioCentury, Korea Advanced Institute of Science and Technology, Daejeon, Republic of Korea
[e]Department of Molecular Microbiology and Bioenergetics, Institute of Molecular Biosciences, Johann Wolfgang Goethe University Frankfurt/Main, Frankfurt, Germany
[f]Intelligent Synthetic Biology Centre, Daejeon, Republic of Korea

**ABSTRACT** Acetogens synthesize acetyl-CoA via the $CO_2$-fixing Wood-Ljungdahl pathway. Despite their ecological and biotechnological importance, their translational regulation of carbon and energy metabolisms remains unclear. Here, we report how carbon and energy metabolisms in the model acetogen *Acetobacterium woodii* are translationally controlled under different growth conditions. Data integration of genome-scale transcriptomic and translatomic analyses revealed that the acetogenesis genes, including those of the Wood-Ljungdahl pathway and energy metabolism, showed changes in translational efficiency under autotrophic growth conditions. In particular, genes encoding the Wood-Ljungdahl pathway are translated at similar levels to achieve efficient acetogenesis activity under autotrophic growth conditions, whereas genes encoding the carbonyl branch present increased translation levels in comparison to those for the methyl branch under heterotrophic growth conditions. The translation efficiency of genes in the pathways is differentially regulated by 5′ untranslated regions and ribosome-binding sequences under different growth conditions. Our findings provide potential strategies to optimize the metabolism of syngas-fermenting acetogenic bacteria for better productivity.

**IMPORTANCE** Acetogens are capable of reducing $CO_2$ to multicarbon compounds (e.g., ethanol or 2,3-butanediol) via the Wood-Ljungdahl pathway. Given that protein synthesis in bacteria is highly energy consuming, acetogens living at the thermodynamic limit of life are inevitably under translation control. Here, we dissect the translational regulation of carbon and energy metabolisms in the model acetogen *Acetobacterium woodii* under heterotrophic and autotrophic growth conditions. The latter may be experienced when acetogen is used as a cell factory that synthesizes products from $CO_2$ during the gas fermentation process. We found that the methyl and carbonyl branches of the Wood-Ljungdahl pathway are activated at similar translation levels during autotrophic growth. Translation is mainly regulated by the 5′-untranslated-region structure and ribosome-binding-site sequence. This work reveals novel translational regulation for coping with autotrophic growth conditions and provides the systematic data set, including the transcriptome, translatome, and promoter/5′-untranslated-region bioparts.

**KEYWORDS** acetogen, acetogenesis, translational regulation, Wood-Ljungdahl pathway

Acetogenesis has received immense attention because it offers an efficient route for microbial metabolism to convert one-carbon ($C_1$) gaseous feedstocks, such as carbon monoxide (CO) and carbon dioxide ($CO_2$), to acetyl-CoA via the reductive

Address correspondence to Byung-Kwan Cho, bcho@kaist.ac.kr.

acetyl-CoA pathway, known as the Wood-Ljungdahl (WL) pathway (1, 2). This unique pathway is presumed to be the first carbon fixation pathway on earth (3) and enables acetogens to grow autotrophically using $C_1$ gases as the sole carbon sources (4, 5). A few system-level analyses for acetogens revealed that the expression of acetogenesis-related genes was upregulated under autotrophic growth conditions (6–10); however, the corresponding protein abundances presented low correlations with the mRNA levels (7). While the transcriptional regulation of acetogenesis-related genes is well described, the mechanism by which acetogens consistently maintain efficiently balanced translation levels under autotrophic growth conditions remained unclear (11). In general, protein synthesis in bacteria is highly energy consuming, using up approximately half of the energy utilized for cell growth (12). Since acetogens grow at the thermodynamic limit of life (13), translational regulation is inevitable, and thus, resource allocation should be optimized (14). Therefore, we speculated that the synthesis of essential proteins for autotrophic growth is tightly regulated and that the balanced expression of acetogenesis-related proteins should be crucial for efficient metabolic reactions under autotrophic growth conditions.

Here, we provide the first systematic analysis of gene expression in *Acetobacterium woodii*, one of the model acetogens, at both the transcriptional and translational levels under heterotrophic (fructose) and autotrophic ($H_2$ plus $CO_2$ [$H_2$+$CO_2$]) growth conditions. Based on multi-omics analysis, we found that *A. woodii* efficiently controls the translation of acetogenesis-related proteins under autotrophic growth conditions. Importantly, the genes encoding the methyl and carbonyl branches are translated at similar levels during autotrophic growth, whereas the translation levels of the genes from the carbonyl branch are increased compared to those of genes from the methyl branch under heterotrophic growth conditions. In addition, promoter and 5'-untranslated-region (5'-UTR) analyses suggest that the translation efficiency is differentially regulated by 5'-UTR and ribosome-binding-site (RBS) sequences under autotrophic growth conditions. Our findings provide a crucial basis for understanding acetogenesis from organic and inorganic substrates and further engineering acetogens to produce value-added products from $C_1$ gaseous feedstocks with better productivity.

## RESULTS

**Genome-scale determination of translation levels.** Ribosome profiling (Ribo-Seq) enables the monitoring of protein synthesis efficiency at a genome-wide scale by using deep sequencing of ribosome-protected mRNA fragments (RPFs) (15). Changes in the ratio between the RPF and mRNA transcript levels can be used to identify translational regulation under the conditions of interest (15–17). To determine the changes in the translation levels of *A. woodii* genes under autotrophic and heterotrophic conditions, we generated more than 10.3 million Ribo-Seq reads with an average read length of 30 to 32 bp, indicating at least an 87.8× sequencing depth (see Table S1 in the supplemental material). The number of RPF reads per gene was then normalized using DEseq2 (18) (Table S2), resulting in a high degree of correlation between the biological duplicates (Pearson's $r > 0.97$) (Fig. S1). We also obtained 1.2 million to 4.7 million RNA sequencing (RNA-Seq) reads, corresponding to at least 42.8-fold coverage (Table S1). As expected, a positive correlation was observed between transcription and translation levels under heterotrophic (Pearson's $r = 0.86$) and autotrophic (Pearson's $r = 0.90$) growth conditions, respectively (Fig. 1A). For example, the hydrogen-dependent carbon dioxide reductase (HDCR) genes (*fdhF2*, *hydA2*, *fdhD*, *hycB2*, and *hycB3*) were highly upregulated at the transcription (fold changes [FCs] = 5.2 to 10.2; $P$ value adjusted for multiple testing with the Benjamini-Hochberg procedure [$P_{adj}$] of $<8.42 \times 10^{-20}$) and translation (fold changes = 3.0 to 4.5; $P_{adj} < 5.97 \times 10^{-28}$) levels under autotrophic growth conditions (Fig. 1B; see also Text S1 in the supplemental material). However, the genes encoding selenium-free formate dehydrogenase (FDH) (FdhF1) and FdhF1-specific HycB1 were not expressed at both the transcription and translation levels under both growth conditions (Text S1).

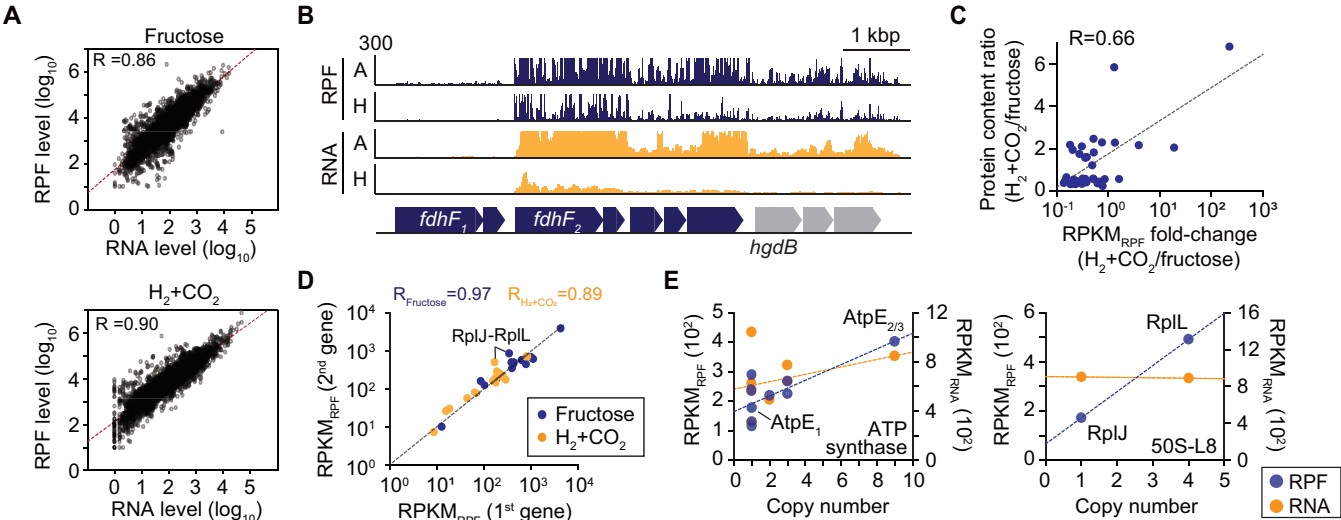

**FIG 1** Determination of the translational landscape of *A. woodii*. (A) Scatterplot of all pairwise $\log_{10}$ expression levels between mRNA (*x* axis) and RPF (*y* axis), presenting a positive correlation. See Tables S1 and S2 in the supplemental material for sequencing statistics and all RPF levels, respectively. (B) Example of RNA-Seq and Ribo-Seq (RPF) profiles for hydrogen-dependent carbon dioxide reductase (HDCR) (Awo_c08190 to Awo_c08260), highlighted in dark blue. H, heterotrophic growth; A, autotrophic growth. (C) Comparison of translation levels to protein abundances. Points represent matching ORFs in the fold changes of translation levels and the changes in the amounts of protein previously reported (20) during growth on fructose versus growth on $H_2+CO_2$ (Pearson's $r = 0.66$). (D) Proportional translation levels for the first and second genes in the operon were compared during growth on fructose or $H_2+CO_2$. (E) Translation levels (RPKM$_{RPF}$) of the $F_1F_o$ ATP synthase operon (Awo_c02150 to Awo_c02240) and the RplJ and RplL ribosomal proteins (Awo_c10870 to Awo_c10880) during autotrophic growth in correlation to their protein copy numbers in the enzyme complexes.

We defined the translation level of each gene as the number of ribosomes engaged in translation elongation per gene length, with correspondence to reads per kilobase per million RPFs (RPKM$_{RPF}$) (10, 15, 17, 19). The comparison between the fold changes of RPKM$_{RPF}$ values and the changes in the amount of protein previously reported in *A. woodii* during growth on fructose versus growth on $H_2+CO_2$ (20) showed a moderate correlation (Pearson's $r = 0.66$) (Fig. 1C). Discrepancies may be attributed to slight differences in culture conditions, the possibility that protein turnover rates or ribosome elongation rates are not constant for all genes, or experimental bias in the process of mass spectrometry analyses.

For validation of data quality, we initially investigated whether subunits of multimeric protein complexes are translated according to their stoichiometry under autotrophic growth conditions (15). For this analysis, we examined the known stoichiometry of the multimeric protein complexes in *A. woodii*. A total of 64 subunits of 17 protein complexes were examined, and the translation levels were linearly correlated between the first and second subunits of the multimeric protein complexes (Pearson's $r = 0.97$) (Fig. 1D). RPKM$_{RPF}$ values of nonequimolar subunits such as RplJ/L and AtpB/E were also proportional to the respective stoichiometries, ranging from 1- to 9-fold, whereas RPKM$_{RNA}$ values were less correlated (Fig. 1E). Thus, the translation of each subunit matched the respective stoichiometry of protein complexes for achieving the correct stoichiometry in multimeric protein complexes.

**Translation levels of acetogenesis-related genes under autotrophic growth conditions.** The transcriptomic results suggest that the genes encoding acetogenesis as well as gluconeogenesis and the pentose phosphate pathway were activated for biomass formation and $CO_2$ reduction during autotrophic growth (6–8, 10) (see Fig. S2 and Text S1 in the supplemental material), as is the case with *Acetobacterium bakii* (9). Furthermore, we investigated the translation levels of the acetogenesis-associated genes under autotrophic growth conditions (Table S2).

First, the HDCR complex revealed dynamic translational regulation under autotrophic growth conditions within a 4-fold range (Fig. 2A). Among the subunits of the HDCR complex, FdhF2 (RPKM$_{RPF}$ = 562.6) and HycB2 (RPKM$_{RPF}$ = 502.6) were translated at rates more than 2.2 times higher than those of HycB3 (RPKM$_{RPF}$ = 228.3) and HydA2 (RPKM$_{RPF}$ = 259.9). The FdhF2 and HycB2 complex also catalyzes the reduction of $CO_2$ to formate with

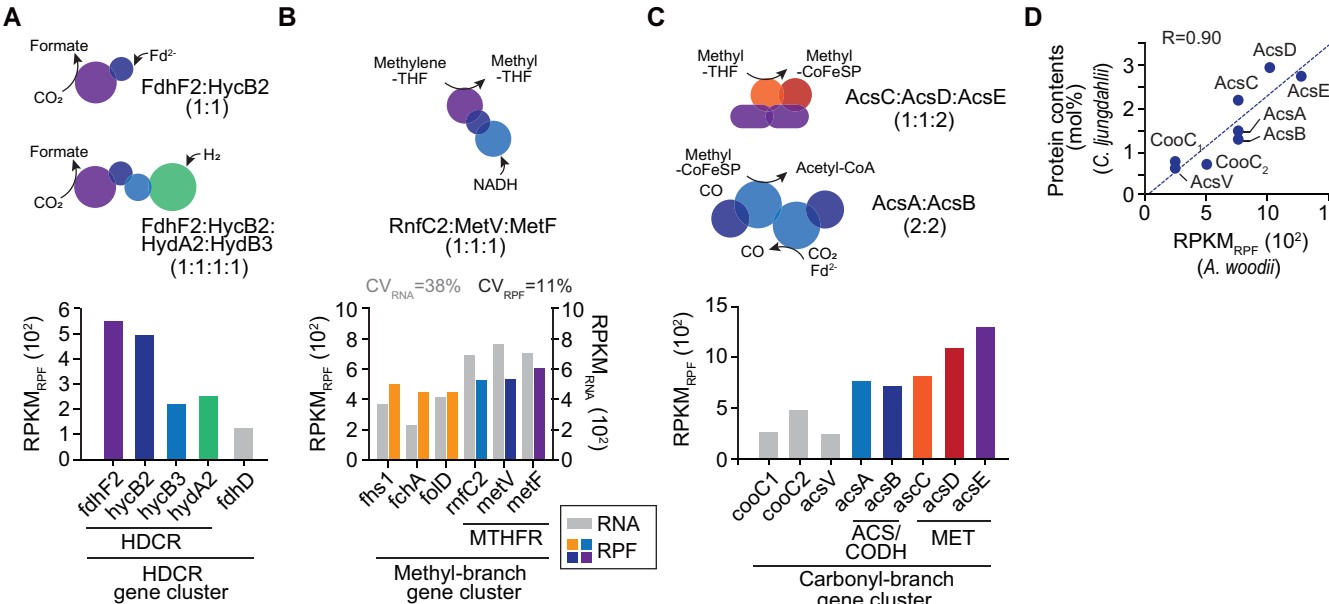

**FIG 2** Translational status of the WL pathway under autotrophic growth conditions. (A) RNA expression levels and translation levels of hydrogen-dependent carbon dioxide reductase (HDCR) under autotrophic growth conditions. Illustrations and subunit ratios of the HDCR complex are shown. For $CO_2$ reduction, electrons are provided by either reduced ferredoxin or the HydA-HydB3 subunits, where $H_2$ oxidation occurs. (B) Comparison between the mRNA ($RPKM_{RNA}$) and translation ($RPKM_{RPF}$) levels of the methyl branch of the WL pathway under autotrophic growth conditions. Illustrations and subunit ratios of the MTHFR complex are shown. The coefficient of variation (CV) (SD of the mean × 100) was calculated based on the $RPKM_{RPF}$ values of all genes in the methyl branch gene cluster under each growth condition. (C) RPF levels for the carbonyl branch of the WL pathway under autotrophic growth conditions. Illustrations and subunit ratios of the ACS/CODH and MET complexes are shown. (D) Correspondence between the published protein abundance of *C. ljungdahlii* (27) and the translation ratio of carbonyl branch proteins. The reported exponentially modified protein abundance index (emPAI) value for each protein was used for determining the protein content (moles percent) using the formula emPAI/Σ(emPAI) × 100. *C. ljungdahlii* values are plotted against the relative translation levels of *A. woodii* (Pearson's $r = 0.90$).

ferredoxin, not using $H_2$ (21), reflecting in particular the protein synthesis rate of the HDCR gene cluster. Six genes in the methyl branch were similarly translated as expected from the subunit stoichiometry (coefficient of variation [CV] = 11%) (Fig. 2B). For example, methylene-tetrahydrofolate (THF) reductase (MTHFR) is a unique heterotrimer complex comprising the MetF, MetV, and RnfC2 subunits (22). They were equally translated according to their subunit stoichiometry ratio (1:1:1) under autotrophic growth conditions. However, genes from the methyl branch were not similarly transcribed (CV = 38%) (Fig. 2B), resulting in significant differences between the $CV_{RNA}$ and $CV_{RPF}$ ($P = 0.03$).

Second, the enzymes in the carbonyl branch were translated in several different ratios under autotrophic growth conditions (Fig. 2C). The subunits of two large protein complexes (220 to 310 kDa), the methyltransferase/corrinoid iron-sulfur protein (MET/CoFeSP) and the acetyl-CoA synthase/carbon monoxide dehydrogenase (ACS/CODH) complexes, had relatively high translation levels (3- to 5-fold higher) than others in the carbonyl branch. The homodimeric MET protein (AcsE) (23) interacts with two CoFeSPs, and each CoFeSP comprises small and large subunits (AcsC and AcsD, respectively) (5, 24). Thus, presumably, a higher translation level of AcsE is required for efficient MET/CoFeSP complex assembly. Similar translation levels (CV = 6.5%) were observed for *acsA* (Awo_c10740) and *acsB* (Awo_c10760), which encode the heterotetrameric ACS/CODH complex (A2B2) (Fig. 2C) (25, 26). We further validated the translation levels of the genes in the carbonyl branch by comparing our data against the protein abundances of *Clostridium ljungdahlii* under syngas fermentation conditions (27). Our results were in accordance with the protein abundance, with Pearson's $r$ value of 0.90 (Fig. 2D).

Third, the subunits of the electron-bifurcating hydrogenase complex ($CV_{RPF} = 24\%$; $RPKM_{PRF} = 107.3$ to $204.9$; $CV_{RNA} = 46\%$; $RPKM_{RNA} = 239.8$ to $917.3$) encoded by *hydA1*, *hydB*, *hydD*, and *hydC* and the Rnf complex ($CV_{RPF} = 23\%$; $RPKM_{PRF} = 285.6$ to $517.8$; $CV_{RNA} = 23\%$; $RPKM_{RNA} = 166.1$ to $317.5$) encoded by *rnfB*, *rnfA*, *rnfE*, *rnfG*, *rnfD*, and *rnfC1* were produced within a 2-fold range (Table S2). These results are consistent with

their subunit stoichiometric relationship. Collectively, the acetogenesis-related genes were effectively regulated at the translational level under autotrophic growth conditions. Moreover, the production of carbonyl branch enzymes was upregulated compared to that of other enzymes related to acetogenesis, reflecting their limited reaction fluxes (16).

**Differential translation levels of acetogenesis-associated genes under heterotrophic and autotrophic growth conditions.** Although the changes in the mRNA and RPF levels of the pentose phosphate pathway, glycolytic/gluconeogenesis pathway, and tricarboxylic acid (TCA) cycle are similar, some of the acetogenesis-related genes were determined to be insignificant from the fructose conditions to the $H_2+CO_2$ conditions (Fig. 3A). In particular, ACS/CODH, MET/CoFeSP, $F_1F_o$ ATP synthase, and the Rnf complex were significantly upregulated at the transcription level (fold changes = 1.8 to 8.5; $P_{adj} < 1.72 \times 10^{-4}$); however, surprisingly, all genes retained a similar ($P_{adj} < 0.85$) or lower RPF level under autotrophic growth conditions (fold changes = 0.68 to 0.89; $P_{adj} < 8.12 \times 10^{-3}$) (see Text S1 in the supplemental material).

Next, the correlation between gene expression and growth conditions across the multimeric protein subunits associated with acetogenesis was investigated at the transcription and translation levels (Fig. 3B). The analysis revealed that the RPF ratios of subunits in multimeric proteins are relatively constant (Pearson's $r = 0.64$ to 1.00) for an individual gene independent of the growth conditions, although the RNA ratio is not correlated (Pearson's $r = -0.40$ to 0.13) for MTHFR, hydrogenase, and the Rnf complex between heterotrophic and autotrophic growth conditions, most likely reflecting tight translational regulation of acetogenesis-associated genes.

Considering that protein production was regulated at both the transcriptional and translational levels, we hypothesized that the key proteins for acetogenesis are more efficiently translated under autotrophic growth conditions. Overall, we observed similar translation levels of genes encoding the HDCR, methyl branch, and carbonyl branch under autotrophic growth conditions ($P > 0.05$ by a Wilcoxon rank sum test), whereas enzymes of the HDCR and carbonyl branch had higher translation levels than the enzymes of the methyl branch under heterotrophic growth conditions ($P < 0.005$ by a Wilcoxon rank sum test) (Fig. 3C). In particular, two large protein complexes (ACS/CODH and MET/CoFeSP) (median $RPKM_{RPF} = 3,833.2$) were upregulated up to 12-fold at the translation level compared to the HDCR complex (median $RPKM_{RPF} = 311.8$) under heterotrophic growth conditions, suggesting unbalanced protein expression among the WL pathway proteins. Furthermore, we observed that $RPKM_{RPF}$ values of $F_1F_o$ ATP synthases and their protein copy numbers were less correlated under heterotrophic growth conditions (Pearson's $r = 0.66$; $P > 0.05$), whereas the $RPKM_{RPF}$ and respective stoichiometries were significantly similar under autotrophic growth conditions (Pearson's $r = 0.80$; $P = 0.01$), indicating the efficient translation of each subunit under $H_2+CO_2$ conditions (Fig. 3D).

To determine whether the unbalanced translation of the WL pathway observed under heterotrophic growth conditions affects acetogenesis, resting cell assays were performed. To test this, we used heterotrophically (H) and autotrophically (A) grown A. woodii cells, resulting in four types of samples (designated H→H, H→A, A→H, and A→A) (Fig. 3E). Interestingly, the levels of acetate production from fructose were similar in autotrophically (A→H, 13.8 mM) and heterotrophically (H→H, 14.4 mM) grown cells. In contrast, acetate production from $H_2+CO_2$ in heterotrophically grown cells (H→A, 7.5 mM) was only about half of that in autotrophically grown cells (A→A, 14.7 mM), suggesting relatively low acetogenesis activity of the heterotrophically grown cells. Based on the data from translational regulation and metabolic activity, we propose that the WL pathway genes are translated at similar levels to achieve efficient acetogenesis activity under autotrophic growth conditions.

**The glycine cleavage system may contribute to heterotrophic acetogenesis.** Since the translation level of carbonyl branch enzymes was higher than that of enzymes from the methyl branch during growth on fructose, we looked for folate-dependent single-carbon reaction sequences that could replenish the carbon in the methyl branch. The intermediates methylene-THF and methyl-THF could be ideal candidates.

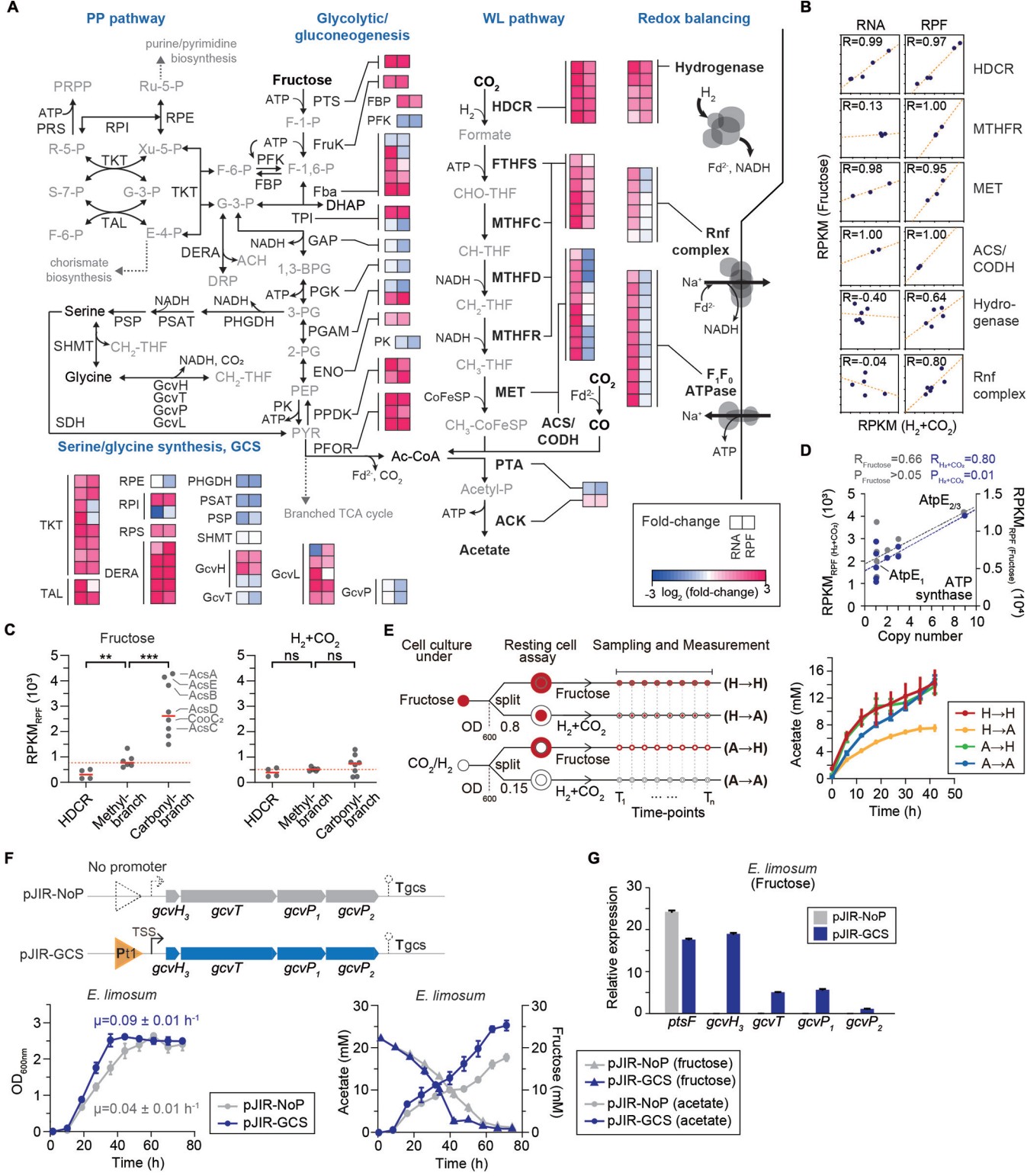

**FIG 3** Differential translation levels of acetogenesis proteins under fructose and $H_2+CO_2$ conditions. (A) Transcriptome and translatome dynamics of central metabolism during heterotrophic and autotrophic growth. Fold changes of genes involved in the pentose phosphate (PP) pathway, glycolytic/gluconeogenesis, the Wood-Ljungdahl (WL) pathway, the branched TCA cycle, the serine/glycine synthesis pathway, the glycine cleavage system (GCS), and redox-balancing systems in *A. woodii* are shown. The heat map shows $log_2$ fold changes of mRNA and RPF between $H_2+CO_2$ and fructose conditions. See Table S2 in the supplemental material for the complete enzyme names and $RPKM_{RPF}$ levels. (B) Correlation of gene expression between the fructose and $H_2+CO_2$ conditions at the transcription ($RPKM_{RNA}$) and translation ($RPKM_{RPF}$) levels. Pearson's correlations (*R*) between the two values are shown. (C) Comparison between the translation levels of WL pathway-related genes under fructose and $H_2+CO_2$ conditions. The vertical scatterplot indicates individual translation levels (black, $RPKM_{RPF}$) and the median value (red line). The red dotted line represents the median value of the methyl branch. The

In the *A. woodii* genome, we found genes encoding the glycine cleavage system (GCS), in which the carbon flow can be coupled to the WL pathway (Fig. 3A and Fig. S3A). Recently, we reported the functional cooperation of the WL pathway and the GCS in *Clostridium drakei* (28), which are functionally interconnected to convert $CO_2$ into acetyl-CoA and acetyl-phosphate. 3-Phosphoglycerate (3-PG), an intermediate in the Embden-Meyerhof-Parnas (EMP) pathway, is converted to serine and glycine via the serine/glycine synthesis pathway. Serine transhydroxymethylase produces methylene-THF and glycine, and the latter is oxidized to $CO_2$ and methylene-THF. Therefore, this system is expected to contribute to the utilization of the carbonyl branch during heterotrophic growth.

Our data revealed that genes in the serine biosynthesis pathway, comprising 3-phosphoglycerate dehydrogenase (PHGDH), phosphoserine aminotransferase (PSAT), and phosphoserine phosphatase (PSP), were significantly upregulated at both the transcription (fold changes = 2.1 to 2.3; $P_{adj} < 0.020$) and translation (fold changes = 1.4 to 2.4; $P_{adj} < 0.017$) levels under heterotrophic growth conditions (Fig. S3B). Meanwhile, glyceraldehyde-3-phosphate dehydrogenase (GAP), phosphoglycerate kinase (PGK), triosephosphate isomerase (TPI), and the GCS gene cluster encoding glycine dehydrogenase (GLDC) (GcvH), aminomethyltransferase (AMT) (GcvT), and glycine cleavage system H protein (GcvH) were significantly upregulated at the translational level only (fold changes = 1.4 to 2.2; $P_{adj} < 1.30 \times 10^{-4}$) during heterotrophic growth. In general, all proteins required for GCS-mediated heterotrophic acetogenesis were significantly activated at the translational level (Fig. S3B). This is in accordance with the hypothesis that the GCS pathway replenishes the methyl branch during growth on fructose.

Since no efficient genetic tools for knockout have been reported yet for *A. woodii*, we investigated the contribution of the GCS to heterotrophic acetogenesis in other acetogens that do not have this system. Most acetogens contain GCS genes, whereas *Eubacterium limosum* lacks key genes of this pathway (Fig. S3C). Considering that the carbonyl branch enzymes of *E. limosum* are at least 2-fold translationally upregulated compared to the enzymes of the methyl branch under heterotrophic growth conditions (10), we assumed that GCS-mediated heterotrophic metabolism with the energy conservation system (29) could be an energetically efficient process in *E. limosum* (Fig. S4A to C). To confirm this, we initially identified the transcription unit of the GCS via transcription start site (TSS) and termination site determinations (Fig. 3F), showing that *gcvHTP* may form an operon. The genes were cloned into plasmid pJIR-GCS under the control of the p*tetO1* (pt1) promoter, and the plasmid was transformed into *E. limosum*. The expression of the *gcvHTP* genes led to a 2.25-fold increase in the growth rate ($P < 0.002$) on fructose compared to the control strain (Fig. 3F and G). Along with the growth rate increase, relatively rapid fructose reduction and 140% higher acetate production (25.3 mM) were also observed in the GCS strain, compared with the control strain. These results indicated that the heterologous expression of the GCS in *E.*

**FIG 3** Legend (Continued)
significance of differences was assessed by the Wilcoxon rank sum test (ns, not significant; **, $P < 0.01$; ***, $P < 0.001$). (D) Correlation between the protein copy numbers of proteins and the corresponding translation levels (RPKM$_{RPF}$) of the $F_1F_o$ ATP synthase operon under fructose and $H_2+CO_2$ conditions. Pearson's correlations (*R*) and *P* values between the two values are shown. (E) Comparison of acetate production from fructose and $H_2+CO_2$ by *A. woodii* cell suspensions. The assay design is presented on the left. Resting cell assays were performed in imidazole buffer with 20 mM NaCl under strictly anoxic conditions (<5 ppm $O_2$) (right). Cell suspensions were prepared from cells grown on fructose or $H_2+CO_2$ and analyzed for their ability to produce acetate from fructose or $H_2+CO_2$ (80:20 [vol/vol]; 200 kPa), resulting in four types of samples (designated H→H, H→A, A→H, and A→A). Acetate production was measured via HPLC, and all values are means ± standard errors of the means (SEM) obtained from three independent experiments. (F, top) Design of plasmid-based expression for the characterization of GCS contribution to the heterotrophic growth of *E. limosum*. Plasmid pJIR-GCS harbors a transcription unit of *gcvTHP* genes under the control of the p*tetO1* (pt1) promoter. Plasmid pJIR-NoP, which harbors *gcvTHP* genes without a promoter (NoP), was also constructed and used as a negative control. (Bottom) Cell growth (left) and metabolite profiles (right) of the GCS and NoP strains under heterotrophic growth conditions. Cells were grown using DSMZ medium 135 supplemented with 4 g liter$^{-1}$ fructose, 2 g liter$^{-1}$ NaCl, 15 $\mu$g ml$^{-1}$ thiamphenicol, and 30 ng ml$^{-1}$ anhydrotetracycline. The values are presented as the means from three different biological replicates ± SEM. (G) Quantitative RT-PCR data for GCS gene expression in the *E. limosum* GCS strain. Plasmid pJIR-NoP, which harbors the *gcvTHP* genes without a promoter, was also constructed and utilized as a negative control. Each value is the mean from three independent replicate experiments, and error bars indicate SEM. The *ptsF* gene (ELIM_c2548) (fructose-specific PTS) was used as a positive control. The housekeeping gene *gyrA* (ELIM_c0629) (DNA gyrase subunit A) was used as the reference.

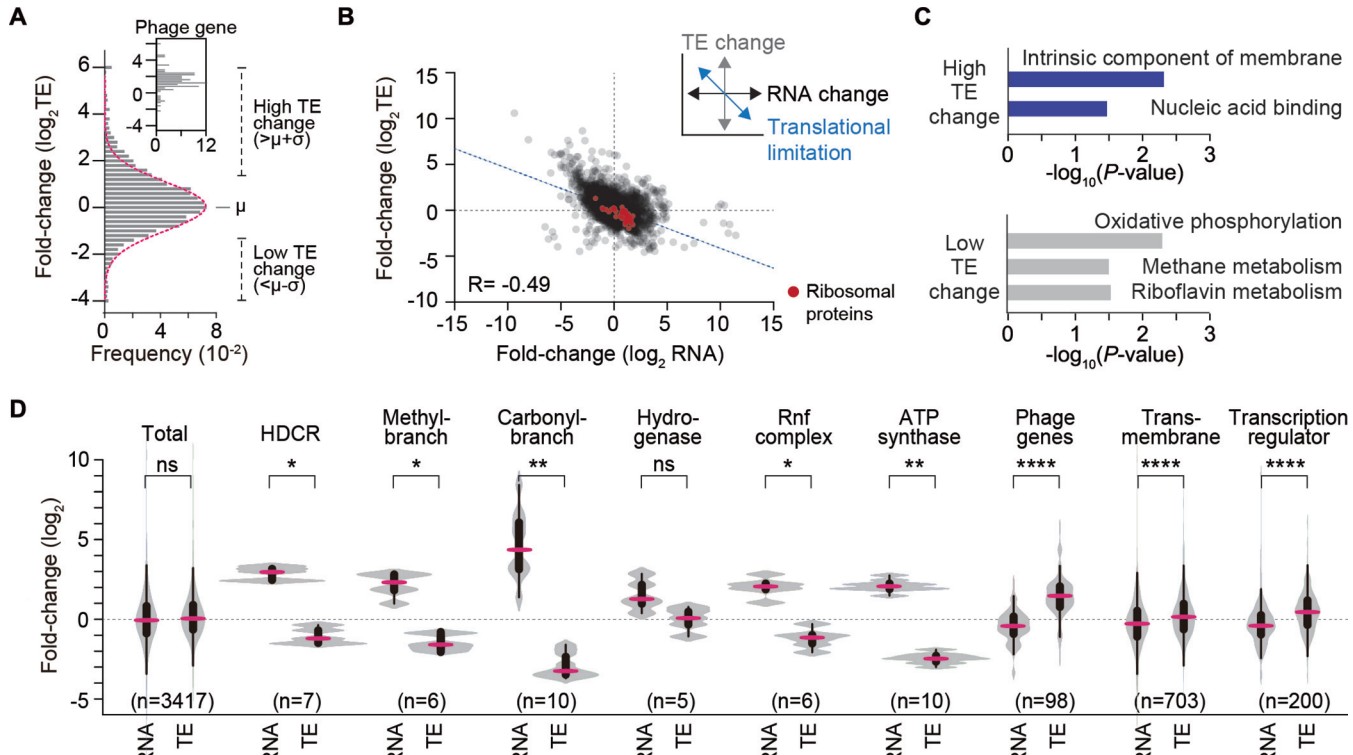

**FIG 4** Global translational dynamics of *A. woodii* between fructose and $H_2+CO_2$ conditions. (A) Distribution of differences in translation efficiency (TE) for all proteins (autotrophic/heterotrophic growth conditions). Most of the phage-related genes showed a high TE under the autotrophic growth conditions (top). The black dotted line indicates the $\log_2$ fold change threshold ($\mu = 0.1$; $\sigma = 1.4$ [$\log_2$ scale]). The red dotted line indicates the Gaussian fit curve. See Table S2 in the supplemental material for all TE values. (B) Fold changes ($\log_2$ scale) in mRNA abundances and TEs (autotrophic/heterotrophic growth conditions), presenting a negative correlation. (C) Enriched GO/KEGG pathways of the translationally regulated genes (Bonferroni-corrected P value of <0.05). (D) mRNA fold changes and translation efficiency fold changes of all genes (autotrophic/heterotrophic growth conditions): hydrogen-dependent carbon dioxide reductase (HDCR) (Awo_c08190 to Awo_c08260), the methyl branch gene cluster (Awo_c09260 to Awo_c09310), the carbonyl branch gene cluster (Awo_c10670 to Awo_c10760), the bifurcating hydrogenase gene cluster (Awo_c26970 to Awo_c27010), the Rnf complex gene cluster (Awo_c22010 to Awo_c22060), $F_1F_0$ ATP synthase (Awo_c02140 to Awo_c02240), phage-related genes, transmembrane genes, and genes encoding putative transcriptional regulators. We analyzed the translational dynamics for phage-related genes, where two large phage-related gene clusters (Awo_c30850 to Awo_c31230 and Awo_c34690 to Awo_c35270) were inspected. See Table S2 for all TE values. To analyze translational regulation for obligate members of transmembrane proteins, we created a list of transmembrane proteins with predicted or known topologies based on data in the UniProt 2017_12 database. Moreover, we obtained a list of regulators that were annotated as "transcriptional regulators" from the UniProt 2017_12 database. Significance was assessed by a two-tailed Wilcoxon signed-rank test (ns, not significant; *, $P < 0.05$; **, $P < 0.01$; ****, $P < 0.0001$).

*limosum* conveys a fitness advantage during growth on fructose. However, an additional knockout study is required to validate the GCS function in *A. woodii*.

**Changes in translation efficiency between heterotrophic and autotrophic growth conditions.** Translation in bacteria is known to be regulated by controlling their quantity of active ribosomes (30) by RNAs or binding of regulatory proteins (31). Considering the dynamic and substrate-specific translational regulation of acetogenesis-associated genes, we hypothesized that differences in translational efficiency (TE) contribute substantially to the control of gene expression in the acetogenesis of *A. woodii*. To address this, we calculated the TE of each gene by dividing the RPF density by the corresponding mRNA level (RPF/mRNA) and examined the changes in TE between the two growth conditions (autotrophic/heterotrophic growth conditions). Most genes revealed a 20-fold range change in the TE (0.2 to 4 $\log_2$). By examining the fold change threshold (1 standard deviation [SD] from the mean at a $\log_2$ scale), we identified 547 high-TE (16%) and 274 low-TE (8%) genes under the two growth conditions (Fig. 4A). A negative correlation between the mRNA change and TE change (Pearson's $r = -0.49$) indicated translational buffering (Fig. 4B) (32). This observation demonstrates that ribosome abundance is limited compared to the transcript level, which is supported by the fact that the TE of ribosomal proteins is downregulated under autotrophic growth conditions. Gene Ontology (GO)/KEGG pathway enrichment

analysis of the translationally regulated genes revealed 27 transcriptional regulators, such as the LambdaCh01/AraC/XRE families, a lactate/cellobiose phosphotransferase system (PTS), and 69 putative membrane proteins upregulated under autotrophic growth conditions (Fig. 4C). Genes involved in oxidative phosphorylation, methane metabolism, and riboflavin metabolism were significantly enriched (Bonferroni-corrected $P$ value of $<0.0009$) in the low-TE group found under autotrophic growth conditions.

Notably, transcripts from the WL pathway and energy conservation genes were increased during growth on $H_2+CO_2$, while the TE was decreased (Fig. 4D). The enriched low-TE genes with the GO term oxidative phosphorylation were associated with the $F_1F_o$ ATP synthase operon and the carbonyl branch of the WL pathway. However, although the WL pathway, Rnf complex, and $F_1F_o$ ATP synthase all had a decreased TE under autotrophic growth conditions (median difference of less than $-4.11$ $\log_2$; $P<0.002$ by a Wilcoxon signed-rank test), the electron-bifurcating hydrogenase revealed upregulated expression patterns at the translational (fold change of $>1.45$; $P_{adj}<1.96^{-6}$) as well as transcriptional levels, resulting in insignificant TE changes. In contrast, genes encoding the transmembrane proteins and the transcriptional regulators increased in TE under autotrophic conditions, while the mRNA level decreased. Therefore, these results provide direct evidence that *A. woodii* regulates the gene expression of specific cellular functions at the translational level, including acetogenesis-related genes under autotrophic conditions.

**Translation initiation plays a crucial role in protein synthesis under autotrophic growth conditions.** The wide range (20-fold) of TEs across the genes raises the question of how translation is regulated under autotrophic growth conditions. Although the understanding of translational regulation is limited in *A. woodii*, we hypothesized that the secondary structure of the 5′ UTR could control translation efficiency. In general, the 5′ UTR is a regulatory component that allows the ribosomal machinery to bind to mRNA and initiate translation. This translation initiation step could be a major determinant in TE, as it is the rate-limiting step for the translation process (33). To examine the effect of the 5′ UTR on TE, we first determined the transcription architecture of the *A. woodii* genome using the differential RNA sequencing (dRNA-Seq) method (34) (Fig. 5A, Fig. S5A to F, and Table S2). A total of 581 TSS positions were analyzed based on the ratio of 5′-end read densities ($>2$-fold) between the RNA 5′-pyrophosphatase-treated ($RPP^+$) and -nontreated ($RPP^-$) libraries. Motif searches upstream of sequences of the TSSs resulted in the TSSs containing purine-rich TSSs along with an extended $-10$ box motif (TATAAT) and the $-35$ box motif (TTGACA). Based on these, we assigned 483 5′ UTRs for 455 genes, including all of the acetogenesis-associated genes (Fig. S5G and H). For example, the HDCR operon can be transcribed starting at *fdhF2*, as concluded from the TSS determination (Fig. 5A and B). Several TSS positions were independently validated by using 5′ rapid amplification of cDNA ends (RACE) and Sanger sequencing (Fig. 5B and Fig. S5C and D). In *A. woodii*, the median 5′-UTR length was 49 nucleotides (nt) (Fig. 5C), and this length distribution is similar to the 5′-UTR length distribution of the psychrotolerant acetogen *Acetobacterium bakii* (median length of 46 nt; $P>0.05$ by a Wilcoxon rank sum test) (9). However, the 5′ UTR of genes related to the WL pathway of *A. woodii* (median length of 38 nt) is significantly shorter than that of *A. bakii* (median length of 97 nt; $P<0.03$ by a Wilcoxon rank sum test), showing that the expression of genes related to the acetogenesis of *A. woodii* is not regulated by the 5′ UTR in response to low temperatures, unlike *A. bakii* (9).

Next, we observed ribosome occupancy in 5′ UTRs. Compared to the open reading frame (ORF) region, the median RPF/RNA ratio was 0.48 for the 5′ UTRs under heterotrophic growth conditions (Fig. 5D), indicating that the ribosome density in the ORF is twice that of the 5′ UTRs. Interestingly, we found a significantly increased RPF/RNA ratio (median, 1.38) in the 5′ UTR in comparison to the ORF region under autotrophic growth conditions ($>2.8$-fold; $P<0.0001$ by a Wilcoxon matched-pairs signed-rank test). This unique pattern is closely related to ribosome pausing, which in certain circumstances triggers translational abandonment via the process of *trans*-translation (35). For example, the 5′ UTR of the *cooC1* gene (Awo_c10670) presented increased

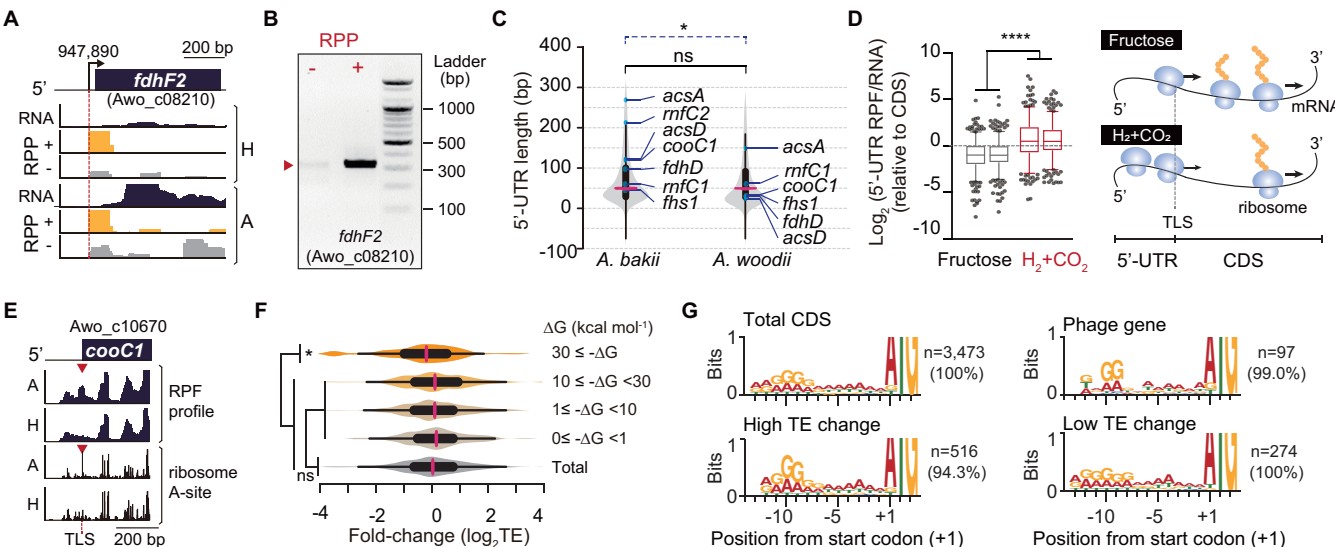

**FIG 5** Analysis of translational regulation with the 5′ UTR. (A) Example of a determined TSS of the HDCR operon and RNA-Seq profiles and dRNA-Seq profiles mapped onto the *A. woodii* genome. RNA 5′-pyrophosphatase-treated (RPP+) and -untreated (RPP−) libraries are represented, and the red dotted line indicates the TSS position. See Table S1 in the supplemental material for sequencing statistics of dRNA-Seq. The total TSSs are listed in Table S2. (B) TSS validation of the HDCR operon using 5′ rapid amplification of cDNA ends (RACE) experiments. The red triangle indicates the expected product length. See Fig. S5 for TSS validation of the WL pathway, Rnf complex operon, the bifurcating hydrogenase operon, and F₁F₀ ATP synthase operon. (C) Comparison of the 5′-UTR length distributions between *A. bakii* (9) and *A. woodii*. Violin plots show medians and quartile ranges of 5′-UTR lengths. The black line represents the statistical difference in the 5′-UTR lengths of total genes, whereas the blue dotted line indicates the statistical difference in the 5′-UTR lengths of genes related to the WL pathway according to the Wilcoxon rank sum test (ns, not significant; *, $P < 0.05$). (D) Ribosome occupancy in 5′ UTRs. (Left) Compared to the coding sequence (CDS), the RPF/RNA ratio was calculated in 5′ UTRs under heterotrophic and autotrophic growth conditions. (Right) Autotrophic growth conditions may cause accumulation of ribosomes at the 5′ UTR of mRNA. (E) Autotrophic growth leads to ribosome stalling at the initiation codon of the *cooC1* gene. The position of the A-site of the stalled ribosome obtained from the Ribo-Seq data is also indicated. Our RPF data also represent *cooC1* gene translation that commences upstream in frame with the canonical AUG. (F) TE change distributions across minimum folding free energies (MFEs) of the 5′ UTR. Structures of a representative set of the 5′ UTRs were predicted using ViennaRNA RNAfold (48). Significance was assessed by a two-tailed Wilcoxon signed-rank test (ns, not significant; *, $P < 0.05$). (G) Conserved ribosome-binding sequence for the total CDS, genes with high TE changes, phage-related genes (most of the phage-related genes showed high TEs), and genes with low TE changes at the upstream region of the initiation codon. The high TE change and phage-related gene sets revealed more obvious canonical AG-rich Shine-Dalgarno motifs than the low TE change gene set.

ribosome occupancy around the initiation codon under autotrophic growth conditions (Fig. 5E). Thus, autotrophic growth conditions may cause the accumulation of ribosomes at the 5′ UTR of the gene, suggesting an impediment of translation initiation. The accumulation of ribosomes at the 5′ UTR is also observed in several *Escherichia coli* genes under carbon limitation (36) and in *Saccharomyces cerevisiae* under oxidative stress (37) and amino acid starvation (15).

Next, we investigated the correlation between the TE and the 5′-UTR sequences, indicating that highly structured 5′ UTRs (less than the lower median ΔG value of −30 kcal mol⁻¹) reduce the TE (15, 16) of the downstream genes ($P = 0.029$ by a Wilcoxon rank sum test) under autotrophic growth conditions (Fig. 5F), as is the case with *E. limosum* (10). The Shine-Dalgarno (SD) sequence is of prime importance in identifying the translation initiation site on bacterial mRNA (38), and a canonical SD motif was observed in the phage and high-TE genes (Fig. 5G). These results show that a highly conserved SD sequence is more important to enhance TE under autotrophic growth conditions than under heterotrophic growth conditions, as seen previously with *C. ljungdahlii* (14). Collectively, translational initiation is more important for protein synthesis under autotrophic growth conditions than under heterotrophic growth conditions. The divergent regulations of translation initiation are due in part to the mRNA 5′-UTR structure and SD sequence.

## DISCUSSION

Here, we provide a systematic analysis of gene expression in *A. woodii* at both the transcriptional and translational levels under heterotrophic and autotrophic conditions. The data from the transcriptomic analysis are in line with previous results in *A. bakii* (9)

showing that acetogenesis-related genes, as well as gluconeogenesis/pentose phosphate pathway genes, were activated for biomass formation and $CO_2$ reduction during autotrophic growth. In addition, the gene expression of glycolysis and an incomplete TCA cycle were not activated under autotrophic conditions. Furthermore, our translatome data enabled us to understand how translation is efficiently regulated under energy-limited conditions. At a genome scale, the translation level of subunits of enzyme complexes matched the determined stoichiometries under autotrophic growth conditions. The translation levels were primarily dependent on the mRNA level under both heterotrophic and autotrophic conditions; however, most of the genes changed their TEs up to 20-fold between the two growth conditions, indicating efficient translational regulation during growth on $H_2+CO_2$ (Fig. 4A). Given that the translational capacity of a cell is reallocated in the face of limited resources, protein synthesis rates of the essential enzymes for autotrophic growth should be optimized to achieve a maximal growth rate (16). Under autotrophic conditions, the TE of ribosome biogenesis decreased, resulting in a greater restriction of the available ribosome protein than of the mRNA. Thus, *A. woodii* was assumed to be carrying out protein synthesis economically, thus focusing on the balanced translation of the acetogenesis-related proteins for the most efficient metabolic reactions. Specifically, these data revealed that the translation levels of the methyl branch and energy conservation enzymes were effectively controlled in similar amounts under autotrophic growth conditions. Interestingly, the carbonyl branch enzymes revealed relatively variable translation levels (from 1- to 5-fold), which is consistent with data from the *C. ljungdahlii* proteome study under syngas conditions (27). In particular, MET/CoFeSP and ACS/CODH show higher translation levels than other WL pathway-related enzymes. This probably compensates for the rate-limiting catalytic step (39) by the large conformational changes during the enzymatic reactions (24). Although the TE of the carbonyl branch was significantly downregulated, it still presented high protein fractions ($\sim$7.8%) of the total protein content (27, 40) under autotrophic growth conditions. Presumably, this represents a strategy for allocating limited protein synthesis capacity, as is the case with *C. ljungdahlii* (14, 41, 42). However, further proteomic analyses will be needed to quantify the abundance of these carbonyl branch enzymes at the protein level in *A. woodii*.

Acetogenesis is a modular metabolism in which the oxidative part is combined with the reductive part by appropriate electron donors such as $H_2$, NADH, and reduced ferredoxin (43). Based on the translatome data, we hypothesized that carbon can also flow from the oxidative part (glycolysis) to the reductive part (WL pathway) by the GCS in *A. woodii*. The GCS can be regarded as an anaplerotic metabolic pathway that fills up the methyl branch of the WL pathway. Under these conditions, the Rnf complex is less important for redox homeostasis, and thus, less ATP is formed. This is compensated for by higher ATP formation in the central pathways. A beneficial effect of the GCS is also evident from the observation that *E. limosum* containing the GCS grew faster and produced more acetate than the parental strain. Thus, our results suggest that the GCS contributes to $CO_2$ fixation and redox balancing with energy conservation under heterotrophic growth conditions. Under autotrophic growth conditions, it was also reported that $CO_2$ can be converted into acetyl-CoA and acetyl-phosphate via the functional cooperation of the WL pathway, GCS, and glycine reductase in *C. drakei* and *E. limosum* (28). However, further analyses are required to validate the function of the GCS in *A. woodii* proposed from our data set.

The RPF read density observed in the 5′ UTRs indicates the presence of the attached 70S ribosome and that ribosome assembly may occur upstream of the SD sequence. The 70S ribosome requires additional sliding to place the ribosome above the start codon. In particular, the high RPF read density at $-10$ to $\sim$15 nt from the start codon indicates that the ribosomes are impediments due to the binding of the ribosome to the SD sequence (see Fig. S1H in the supplemental material). Although the regularity is low, the average RPF read density at the 5′ UTR shows 3-nt periodicity, indicating the stepwise movement of the ribosome. As described in *E. coli* during nutrient limitation (30), the number of

active ribosomes may be reduced through inhibition of IF2-mediated translation initiation by ppGpp, which increases during nutrient limitation (44). The increased ribosome density in the 5′ UTR during autotrophic growth indicates that the ribosome is inactivated within the noncoding region of the mRNA. Such ribosome inactivation can be beneficial to cells in an environment of insufficient energy, as for the regulation of specific metabolic pathways by reducing the overall translation capacity. Our 5′-UTR analysis suggests that translation initiation is more important for protein synthesis during autotrophic growth than during growth on fructose, and it is regulated by the 5′-UTR structure (10) and SD sequence (14). Importantly, autotroph-responsive TSS selection of the *acsD* gene was identified with validated TSSs, resulting in alternative 5′-UTR mRNAs between the fructose and the $H_2+CO_2$ conditions. Although the $H_2+CO_2$ conditions form short 5′ UTRs (31 nt) in *acsD*, the fructose conditions form long 5′ UTRs (121 nt), resulting in a complex secondary structure for the 5′ UTR (Fig. S5B).

Collectively, the systemic approach to determine both the transcriptome and translatome enabled us to unravel the TE, which calculates the actual rate of mRNA translated into proteins, and to identify GCS-coupled heterotrophic acetogenesis as well as translational control of genes encoding the GCS, the WL pathway, and energy conservation. In particular, understanding the regulation of acetogenesis and collecting diverse bioparts derived from multi-omics data are prerequisites for the biotechnological application of acetogens. It may be feasible to improve the translation levels of key enzymes for strain engineering in order to optimize the metabolic process of syngas fermentation or to produce native or nonnative products with better productivity.

## MATERIALS AND METHODS

**Bacterial strains and growth conditions.** *Acetobacterium woodii* DSM 1030 and *Eubacterium limosum* ATCC 8486 were obtained from the Leibniz Institute DSMZ (German Collection of Microorganisms and Cell Cultures). *A. woodii* and *E. limosum* were cultivated strictly anaerobically at 30°C, in DSMZ medium 135 (45) supplemented with 2 g liter$^{-1}$ NaCl. Fructose (5 g liter$^{-1}$) or a gas mixture of $H_2+CO_2$ (80:20; 200 kPa) was used for the heterotrophic and autotrophic growth conditions, respectively. *Escherichia coli* DH5α was obtained from Enzynomics, Inc. (Daejeon, South Korea), and used for plasmid cloning and maintenance.

**Analytical methods.** Growth was monitored by measuring the optical density at 600 nm ($OD_{600}$). To remove cell debris, 500-μl samples were centrifuged at $14,000 \times g$ for 10 min at 4°C. All metabolite samples were prepared by filtration through 0.2-μm Minisart RC15 syringe filters (Sartorius, Goettingen, Germany). Metabolic products were analyzed using a Waters 1525 high-performance liquid chromatography (HPLC) system (Waters, Milford, MA, USA) equipped with a MetaCarb 87H organic acid column (Agilent Technologies, Wilmington, DE, USA) and a refractive index (RI) detector (Waters), operating at 37°C. $H_2SO_4$ (0.01 N) was used as the mobile phase. The flow rate of the mobile phase was 0.6 ml min$^{-1}$, and the volume of each injection was 20 μl.

**RNA extraction and RNA sequencing library preparation.** For RNA extraction, cell cultures of heterotrophically and autotrophically grown cells were harvested at $OD_{600}$s of 0.8 and 0.15, respectively. Cell cultures were harvested by centrifugation at $4,000 \times g$ for 10 min at 4°C and resuspended in 500 μl of lysis buffer (20 mM Tris-HCl [pH 7.4], 140 mM NaCl, 5 mM $MgCl_2$, and 1% Triton X-100). Resuspended cells were flash-frozen in liquid nitrogen and then ground using a mortar and pestle. Unbroken cells and cell debris were removed by centrifugation at $4,000 \times g$ for 10 min at 4°C. Purified total RNA was then obtained using TRIzol reagent (Thermo Fisher Scientific, Waltham, MA, USA) according to the manufacturer's instructions. To remove the remaining genomic DNA, rDNase I (Ambion, Thermo Fisher Scientific) was added to the purified RNA according to the manufacturer's instructions. Total RNA was measured using the Qubit RNA HS assay kit. RNA quality was confirmed by the $A_{260}/A_{280}$ ratio (>1.8) and by visualization of the two distinct bands of rRNAs by 2% agarose gel electrophoresis. rRNAs were specifically removed by the Ribo-Zero rRNA removal kit (bacteria) (Illumina, Inc., San Diego, CA, USA) according to the manufacturer's instructions. Strand-specific RNA sequencing (RNA-Seq) libraries were constructed using the rRNA-depleted RNAs and a TruSeq strand mRNA LT sample prep kit (Illumina, Inc.) and then sequenced in an Illumina MiSeq instrument with a 151-bp single-end sequencing recipe.

**Differential RNA-Seq library preparation.** Differential RNA-Seq (dRNA-Seq) libraries were prepared as described previously (34), with some modifications (9). Briefly, 400 ng of the rRNA-depleted RNA was divided into two samples for constructing two different libraries (RPP$^+$ and RPP$^-$). For preparing the RPP$^+$ libraries, the sample containing primary (5′-PPP) and processed (5′-P/5′-OH) transcripts was treated with RNA 5′-polyphosphatase (RPP; Epicentre Biotechnologies, Madison, WI, USA). For the preparation of RPP$^-$ libraries, RPP was omitted from the enrichment step. 5′-RNA adaptors (see Table S3 in the supplemental material) were ligated with T4 RNA ligase (Thermo Fisher Scientific) to the RPP$^+$ and RPP$^-$ samples with a 1:3 molar ratio of RNA to 5′-RNA adaptor. The ligated products were then purified with Agencourt AMPure XP beads (Beckman Coulter, Brea, CA, USA). For cDNA synthesis, random nonamer adaptors (Table S3) and a SuperScript III reverse transcriptase kit (Invitrogen, Carlsbad, CA, USA) were

used, according to the manufacturer's directions. The cDNA libraries were purified with $0.8\times$ volumes of AMPure XP beads and then amplified with the indexed primer (Table S3) and Phusion high-fidelity DNA polymerase (Thermo Fisher Scientific). Amplification was monitored with SYBR green gel stain solution (Invitrogen) on a CFX96 real-time PCR detection system (Bio-Rad Laboratories, Hercules, CA, USA). The PCR conditions were as follows: 98°C for 30 s and several cycles (until the reaction reached the plateau phase) of 98°C for 10 s, 56°C for 20 s, and 72°C for 20 s, followed by 72°C for 5 min. The dRNA-Seq libraries were purified with $0.8\times$ volumes of AMPure XP beads and then sequenced using a 100-bp read recipe on an Illumina HiSeq2500 instrument (Illumina, Inc.). Two biological replicates were sequenced, and the required oligonucleotide sequences are listed in Table S3.

**Ribosome profiling library preparation.** Bacterial cells were pretreated at 30°C for 10 min with 100 $\mu$M chloramphenicol (CM) prior to harvesting by centrifugation at $4,000 \times g$ for 10 min at 4°C. The collected cells were washed with 500 $\mu$l polysome buffer (20 mM Tris-HCl [pH 7.4], 140 mM NaCl, 5 mM MgCl$_2$, and 100 $\mu$M CM) and resuspended in ice-cold lysis buffer (20 mM Tris-HCl [pH 7.4], 140 mM NaCl, 5 mM MgCl$_2$, 100 $\mu$M CM, and 1% Triton X-100). The resuspended cells were flash-frozen in liquid nitrogen. Subsequently, the frozen samples were ground by a mortar and pestle. The cell lysates were recovered by centrifugation at $16,000 \times g$ for 10 min at 4°C, and the supernatant was then transferred to a new tube. To degrade RNA that was not protected by the ribosome, the supernatant was subjected to micrococcal nuclease (MNase; New England BioLabs, Ipswich, MA, USA) digestion using 400 U of MNase. Digestion proceeded for 2 h at 37°C with gentle rotation, and the reaction was stopped by the addition of 10 $\mu$l of EGTA (0.5 M). The enriched monosome fraction was confirmed by visualization of the ribosome-protected mRNA footprints, with sizes ranging from 26 to 34 nt, in a 15% polyacrylamide Tris-borate-EDTA (TBE)-urea gel (Invitrogen). The monosome fraction was specifically recovered using a Microspin S-400 HR instrument (GE Healthcare Life Sciences, Marlborough, MA, USA) according to the manufacturer's instructions. The recovered ribosome-bound RNA was isolated using the TRIzol reagent (Thermo Fisher Scientific), and the remaining rRNAs were removed by the Ribo-Zero rRNA removal kit (bacteria) (Illumina, Inc.) according to the manufacturer's instructions. The footprints were denatured at 80°C for 90 s and then equilibrated to 37°C. Subsequently, footprint samples were incubated at 37°C for 1 h with a solution containing 10 U of T4 polynucleotide kinase (PNK; New England BioLabs), 5 $\mu$l of 10$\times$ T4 PNK buffer (New England Biolabs), 0.5 mM ATP, and 20 U of SUPERase In RNase inhibitor (Thermo Fisher Scientific). Phosphorylated footprint samples were purified using the RNeasy MinElute column (Qiagen, Valencia, CA, USA), and the RNA concentration was evaluated using the Qubit RNA HS assay kit (Invitrogen). Ribosomal profiling (Ribo-Seq) libraries were constructed using the NEBNext small RNA library prep kit for Illumina (New England BioLabs) according to the manufacturer's instructions, and the constructed libraries were sequenced using the 50-bp read recipe on the Illumina HiSeq2500 instrument.

**Resting cell assay.** For the resting cell assay, heterotrophically and autotrophically grown *A. woodii* cells were harvested at $OD_{600}$s of 0.8 and 0.15, respectively. Cell suspensions were prepared as described previously (21), with some modifications. All the following steps were carried out under strictly anoxic conditions (<5 ppm O$_2$) in an anaerobic chamber (Coy Laboratory Products, Grass Lake, MI, USA) filled with 96% N$_2$ and 4% H$_2$. The cell samples were washed twice using imidazole buffer (50 mM imidazole, 20 mM KCl, 20 mM MgSO$_4$, 4.4 $\mu$M resazurin, 4 mM 1,4-dithioerythritol [pH 7.0]) and resuspended in the same buffer. Some of the suspension fractions were aliquoted by centrifugation at $6,000 \times g$ for 10 min at 4°C, and total protein contents were quantified as described above. For the cell suspension experiments, whole cells were resuspended to a protein concentration of 1 mg ml$^{-1}$ in a 100-ml anoxic glass bottle. In the assay, 20 mM NaCl was added to the sample, adjusting the final volume to 20 ml with buffer, and the bottles were incubated at 30°C in a shaking incubator. The headspace pressure was adjusted to 200 kPa with a gas mixture of H$_2$+CO$_2$ (80:20 [vol/vol]; 200 kPa) for autotrophic conditions, or 5 g liter$^{-1}$ fructose was added under 200 kPa N$_2$ for heterotrophic conditions. Acetate was measured using a Waters 1525 HPLC system equipped with a MetaCarb 87H organic acid column (Agilent Technologies) as mentioned above. All values are means obtained from three independent experiments.

**5′ rapid amplification of cDNA ends.** The 5′ rapid amplification of cDNA ends (RACE) analysis was conducted as described previously (9). The validated TSSs and their expected amplicons are shown in Fig. S5. The amplicons were visualized using a 2% agarose gel. The oligonucleotides used for 5′-tag-cDNA library construction are available in Table S3.

**Quantitative RT-PCR assay.** To valid RNA-Seq data independently, quantitative reverse transcription-PCR (RT-PCR) was performed using the RNA extracted from cultures grown under equal growth conditions. The purified RNA samples were converted to cDNA using the SuperScript III first-strand synthesis system (Invitrogen) according to the manufacturer's directions. In total, seven genes related to the WL pathway for constitutive expression were selected to validate the RNA-Seq results. Relative expression was measured using a Kapa SYBR Fast universal quantitative PCR (qPCR) kit (Kapa Biosystems, Wilmington, MA, USA) according to the manufacturer's instructions. All primer sequences were determined using NCBI Primer-BLAST and are listed in Table S3.

**Plasmid construction.** The pJIR750ai plasmid was purchased from Sigma-Aldrich. The TargeTron gene knockout system was removed from pJIR750ai by digestion with PvuI, and the plasmid was then religated using T4 DNA ligase, resulting in a new pJIR plasmid. To construct the pJIR-GCS plasmid, a DNA fragment of the p*tetO1* (pt1) promoter was amplified using primer pair tetO1_F and tetO1_R and then inserted into the BamHI and PvuI sites of the pJIR vector, resulting in pJIR-Pt1. The DNA fragment containing the *tetR* gene from the pdCas9 plasmid (Addgene) was amplified using primer pair ptet_tetR_F and ptet_tetR_R and then inserted into the SacI and BamHI sites in pJIR-Pt1, resulting in pJIR-R-Pt1. Following construction, *gcvHTP* genes from *A. woodii* genomic DNA were amplified using two primer pairs (GCS_IF_F and GCS_IF_R, and GCS_IF_NoP_F and GCS_IF_NoP_R) and then cloned into

pJIR-R-Pt1 using the NcoI and PvuI sites and into pJIR using the PvuI site, resulting in the pJIR-GCS and pJIR-NoP vectors, respectively. These cloning steps proceeded using the In-Fusion HD cloning kit, and all high-fidelity PCR amplifications were carried out using Phusion high-fidelity DNA polymerase (Thermo Fisher Scientific). Purification of PCR products and plasmids from *E. coli* DH5$\alpha$ was performed with the QIAquick PCR purification kit (Qiagen) and the DNA-spin plasmid DNA purification kit (iNtRON Biotechnology, Daejeon, South Korea), respectively. All recombinant plasmids were transformed into *E. coli* DH5$\alpha$ competent cells (Enzynomics, Daejeon, South Korea) for cloning and maintenance. All oligo-nucleotides used for the construction of plasmids are listed in Table S3.

**Deep-sequencing data analysis.** The sequence reads generated from RNA-Seq (150-bp single-end reads), dRNA-Seq (100-bp single-end reads), and Ribo-Seq (50-bp single-end reads) were demultiplexed and trimmed using bcl2fastq v1.8.4. Adaptor sequences and low-quality bases were additionally trimmed using CLC Genomics Workbench 6.5.1 (Qiagen). The trimmed reads were mapped to the reference genome (the genome sequence and annotation were obtained from GenBank accession number CP002987 for *A. woodii* DSM 1030) using CLC Genomics Workbench with the following parameters: mismatch cost of 2, insertion cost of 3, deletion cost of 3, length fraction of 0.9, and similarity fraction of 0.9. For Ribo-Seq data, rRNA and tRNA reads were removed by alignment to rRNA and tRNA regions. rRNA- and tRNA-free reads were aligned to the reference genome with the same parameters as the ones mentioned above.

For RNA-Seq and Ribo-Seq, a gene-wise read count was generated from the trimmed reads using the "bedtools -multicov –bam input.bam –bed input.bed" command. Considering Ribo-Seq data, raw reads were counted for reads mapping to the ORFs, excluding the first and last 5 codons to eliminate the effects of initiation and termination (15, 19). The expression values of the genes were normalized using Bioconductor package DEseq2 (18) and the gene-wise read count table as the input. To identify the differentially expressed genes, the significance levels were set at an adjusted *P* value ($P_{adj}$) of <0.01 and a log$_2$ fold change (log$_2$ FC) of greater than |1|. The translation level (protein synthesis rate) was defined by the RPKM (10, 15, 17, 19). All RPKM values were calculated as RPKM = ($10^9 \times$ number of raw reads mapped to the gene)/(total mapped reads $\times$ length of the gene). Translational efficiencies were calculated by dividing the RPF level by the estimated mRNA level in arbitrary units. Additionally, the 5' end of the RPF was used to estimate the information on ribosomal positions. To define the position of the ribosome A-site, Ribo-Seq was additionally performed as described in a previous report (46). For this, each RPF was estimated by adjusting 17 nt from the 5' end of the RPF reads (15).

dRNA-Seq data were analyzed as described in a previous report (9). Briefly, we performed normalization to the 5' ends of the aligned dRNA-Seq read profiles with a scaling factor, and the ratio between RPP-treated (RPP$^+$) and -untreated (RPP$^-$) reads was calculated for each genomic position. The TSSs were selected from RPP$^+$/RPP$^-$ ratios of peak densities of >2. These TSSs were manually curated and categorized based on their genomic position. TSSs having the highest peak density within a distance <300 bp upstream and 100 bp downstream of annotated genes were categorized as primary TSSs (P), and the others were categorized as secondary TSSs (S). TSSs that were located on the same strand as an annotation or on its opposite strand were classified as internal TSSs (I) and antisense TSSs (A), respectively. If they were not classified into any of the categories mentioned above, they were classified as intergenic TSSs (N). The TSSs were compared with two conditional libraries and were summed to give the total TSSs within the range of $\pm$4 bp. All TSSs are summarized in Table S2. Multilayered RNA-Seq, dRNA-Seq, and Ribo-Seq mapped reads were used to construct genome-wide coverage maps, and all maps were visualized using SignalMap (Roche NimbleGen, Inc., Madison, WI, USA).

**Motif discovery in the promoter.** For motif analysis in promoters, we extracted 50 nt of DNA sequence upstream of each detected TSS position. The conserved $-10$ and $-35$ boxes were analyzed by MEME v4.11.1 (47) (a MEME *P* value cutoff of 0.05 was used).

**Statistical testing.** Data statistical testing (Pearson's correlation coefficient, two-tailed Student's *t* test, two-tailed Wilcoxon signed-rank test, and two-tailed Wilcoxon-Mann-Whitney test) was performed using GraphPad Prism v8 software (GraphPad, San Diego, CA, USA). *P* values of <0.05 were considered statistically significant.

**Data availability.** All transcriptome and translatome raw data in FASTQ format are available in the European Nucleotide Archive under study accession number PRJEB33460.

## SUPPLEMENTAL MATERIAL

Supplemental material is available online only.

**TEXT S1**, PDF file, 0.2 MB.
**FIG S1**, PDF file, 0.6 MB.
**FIG S2**, PDF file, 2.6 MB.
**FIG S3**, PDF file, 0.5 MB.
**FIG S4**, PDF file, 0.6 MB.
**FIG S5**, PDF file, 1.4 MB.
**TABLE S1**, PDF file, 0.1 MB.
**TABLE S2**, XLSX file, 0.8 MB.
**TABLE S3**, PDF file, 0.1 MB.

## ACKNOWLEDGMENTS

This work was supported by the C1 Gas Refinery Program (2018M3D3A1A01055733 to B.-K.C.) through the National Research Foundation of Korea (NRF) funded by the Ministry of Science and ICT. Funding for the open-access charge was provided by the C1 Gas Refinery Research Centre.

B.-K.C. conceived and supervised the study. J.S., S.C., and B.-K.C. designed the experiments. J.S., Y.S., S.K., S.J., and S.C. performed the experiments. J.S., Y.S., S.C., V.M., and B.-K.C. analyzed the data. J.S., J.-K.L., D.R.K., S.K., S.C., V.M., and B.-K.C. wrote the manuscript. All authors read and approved the final manuscript.

We declare no competing interests.

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
