## [Reviewer comments · mSystems]

Genome-scale analysis of *Acetobacterium woodii* identifies translational regulation of acetogenesis

Byung-Kwan Cho, Jongoh Shin, Yoseb Song, Seulgi Kang, Sangrak Jin, Jung-Kul Lee, Dong Rip Kim, Suhyung Cho, and Volker Müller

Corresponding Author(s): Byung-Kwan Cho, Korea Advanced Institute of Science and Technology

Review Timeline:

Submission Date:	June 7, 2021
Editorial Decision:	July 3, 2021
Revision Received:	July 9, 2021
Accepted:	July 9, 2021

Editor: Marnix Medema

Reviewer(s): Disclosure of reviewer identity is with reference to reviewer comments included in decision letter(s). The following individuals involved in review of your submission have agreed to reveal their identity: Nico J Claassens (Reviewer #1)

Transaction Report:

DOI: <https://doi.org/10.1128/mSystems.00696-21>

July 3, 2021

Prof. Byung-Kwan Cho
Korea Advanced Institute of Science and Technology
Department of Biological Sciences
291 Daehak-ro, Yuseong-gu
Daejeon 305-701
Korea (South), Republic of

Re: mSystems00696-21 (Genome-scale analysis of *Acetobacterium woodii* identifies translational regulation of acetogenesis)

Dear Prof. Byung-Kwan Cho:

Thank you for submitting your manuscript to mSystems. We have completed our review and I am pleased to inform you that, in principle, we expect to accept it for publication in mSystems. However, acceptance will not be final until you have adequately addressed a few final reviewer comments with textual suggestions and suggestions for improving the figures. Please note that the labeling suggestion by reviewer #1 is facultative, and performing this experiment will not be required for acceptance. You can make your own decision on whether or not you want to perform this.

If the revision is adequate, I expect to be able to handle it quickly without sending it back to review.

Preparing Revision Guidelines

For complete guidelines on revision requirements for your article type, please see the journal Article Types requirement at <https://journals.asm.org/journal/mSystems/article-types>. **Submissions of a paper that does not conform to mSystems guidelines will delay acceptance of your manuscript.**

Sincerely,

Marnix Medema

Editor, mSystems

Journals Department
Reviewer comments:

Reviewer #1 (Comments for the Author):

The revision in this work have clearly improved the work and I am happy the authors toned down some of their conclusions as I suggested.

Still I have two things that I would suggest to tackle and another remark on one of my previous feedback points that was apparently not clear to the authors.

- Based on my comment that a quantitative proteome analysis is lacking the authors decide to show correlation by translome data in this study with proteome data from an older study from Poehlein et al. However, The author now included a comparison to an older (comparative proteome dataset (Poehlein et al., 2012)). However, from Line 108 and the figure legend on Figure 1C it's totally not clear to me how this was done. From the legend they talk only about the first and second gene in an operon, which operon, which genes? I think this study needs a linear regression analysis for all proteins detected in the proteome data set versus the same proteins in the study from Poehlein (also refer to Poehlein's study in the sentences and legend)! Overall I think it's still a missed opportunity the authors did not perform proteome analysis, I think they should indicate this weakness in their manuscript.

- The others revised figure 2 with adding the protein complexes to better show complex subunit stoichiometries. However, I suggest the authors use unique colours per different type of subunit in the complex. Now several different subunits are all purple or blue, and it's still hard to connect it to the bar diagram with RPF data.

-

- Also the authors commented on my labelling experiment suggestion ((Q3)) that they did not get it. So I will try to explain this better. I suggest the authors do grow *A. woodii* and the *E. limosum* (with and without overexpressed GCV) with labelled $^{13}\text{CO}_2$ and ^{12}C fructose under heterotrophic conditions. If the methylene and formyl-THF are mostly produced through glycine cleavage these

C1-moieties will come from fructose and hence (easy to measure) proteinogenic amino acids which have C1-precursors will be fully unlabelled. If these C1-carbons are coming from the methyl-branch of the Wood-Ljungdahl pathway they will lead to incorporating labelled $^{13}\text{CO}_2$ and hence once labelled methionine and histidine. E.g. Histidine obtains one carbon from formyl-THF (in the biosynthesis of ATP, which is incorporated in histidine, sorry I mentioned wrongly before it come from methylene-THF, but this does not matter, it's the same branch). Another possible amino acid to look at is methionine, which has a carbon coming from methylene-THF. I am not sure if the authors will manage to do such an experiment and it's not essential, but it could be very informative.

Reviewer #2 (Comments for the Author):

Title: Genome-scale analysis of *Acetobacterium woodii* identifies translational regulation of acetogenesis.

Claim: "Our findings provide potential strategies to optimize the metabolism of syngas fermenting acetogenic bacteria for better productivity."

Importance: "This work reveals novel translational regulation for coping with autotrophic growth conditions and provides the systematic dataset including transcriptome, translato~~me~~, and promoter/5'-untranslated region bioparts."

RESULTS

Genome-scale determination of translation levels.

P5, L96: "As expected, a positive correlation was observed...". Why was this to be expected?

P8, L158-162: If the data found in the parentheses is in Table S2, I'm not sure these values are needed here. Or just include the RPKM_{rna} values for each and reference Table S2 for the coefficient of variation.

P9, L172: "genes were remained to be insignificant." Does the author mean "genes were determined to be insignificant?"

P10, L191: "two large protein complexes." Which two large protein complexes were upregulated?

P11, L214: "reaction sequences?" Not sure what is meant by reaction sequences. Please clarify.

DISCUSSION

P19, L391: 'ribosome assemble' should read 'ribosome assembly?'

P18, L381-389: The authors mention two GCS-related genes *lpdA1* and *gcvH1* and how they were upregulated under autotrophic growth conditions. The authors describe how these genes are highly conserved in a specific set of acetogens, yet recent work has shown that these two genes are not highly conserved within the *Acetobacterium* genus (Ross et al., 2019-Defining genomic and predicted metabolic features of the *Acetobacterium* genus). Please include findings from this study

in light of the findings of Poehlein and coworkers, or remove the sentence pertaining to the conservation of these genes, as it is misleading as written.

METHODS

The authors argue that autotrophic growth is energy limited. DSMZ 135 medium contains 2 g/L yeast extract. Does this change the available 'energy' in the system? What happens when less YE is utilized?

Point-by-point responses to the reviewer's comments

Reviewer comments:

Reviewer #1

Comments: The revision in this work has clearly improved the work and I am happy the authors toned down some of their conclusions as I suggested. Still, I have two things that I would suggest to tackle and another remark on one of my previous feedback points that were apparently not clear to the authors.

Response: We thank the reviewer for reading our manuscript critically and framing these insightful comments. As stated by the reviewer, we provide our responses as follows.

Q1. Based on my comment that a quantitative proteome analysis is lacking the authors decide to show correlation by translome data in this study with proteome data from an older study from Poehlein et al. However, the authors now included a comparison to an older comparative proteome dataset (Poehlein et al., 2012). However, from Line 108 and the figure legend on Figure 1C, it's totally not clear to me how this was done. From the legend they talk only about the first and second gene in an operon, which operon, which genes? I think this study needs a linear regression analysis for all proteins detected in the proteome data set versus the same proteins in the study from Poehlein (also refer to Poehlein's study in the sentences and legend)! Overall, I think it's still a missed opportunity the authors did not perform proteome analysis, I think they should indicate this weakness in their manuscript.

Response: Fig. 1C shows the linear regression analysis for all proteins detected in the previous proteome dataset (Poehlein et al., 2012) vs. the corresponding proteins in our study (Pearson's $r = 0.66$). As the reviewer pointed out, we modified the figure legend correctly. Also, we have added the sentence describing the weakness raised by reviewer 1 in the Discussion section.

FIG 1. Determination of the translational landscape of *A. woodii*. (A) A scatter plot of all pairwise \log_{10} expression levels between mRNA (x-axis) and RPF (y-axis), presenting a positive correlation. See Table S1 and Table S2 for sequencing statistics and all RPF levels,

respectively. (B) An example of RNA-Seq and Ribo-Seq (RPF) profiles for hydrogen-dependent carbon dioxide reductase (HDCR, Awo_c08190–Awo_c08260), highlighted in dark blue. H, heterotrophic growth; A, autotrophic growth. (C) Comparison of translation level to protein abundance. Point represent matching ORFs in the fold-change of translation level and the changes in the amount of protein previously reported (20) during growth on fructose *versus* growth on H₂+CO₂ (Pearson’s r = 0.66). (D) Proportional translation for the first and second genes in the operon was compared during growth on fructose or H₂+CO₂. (E) Translation levels (RPKM_{RPF}) of F1FO ATP synthase operon (Awo_c02150–Awo_c02240) and RplJ and RplL ribosomal proteins (Awo_c10870–Awo_c10880) during autotrophic growth in correlation to their protein copy numbers in the enzyme complexes.

(Page 17, Line 361): Interestingly, the carbonyl-branch enzymes revealed relatively variable translation levels (from 1- to 5-fold), which is consistent with the *C. ljungdahlii* proteome study under the syngas condition (27). In particular, MET/CoFeSP and ACS/CODH show higher translation levels than other WL pathway-related enzymes. This probably compensates for the rate-limiting catalytic step (39) by the large conformational changes during the enzymatic reactions (24). Although the TE of the carbonyl-branch was significantly downregulated, it still presented high protein fractions (~7.8%) of the total protein content (27, 40) under autotrophic growth conditions. Presumably, this represents a strategy for allocating limited protein synthesis capacity, as is the case with *C. ljungdahlii* (14, 41, 42). However, further proteomic analyses will be needed to quantify the abundance of these carbonyl-branch enzymes at the protein level in *A. woodii*.

Q2. The others revised Figure 2 with adding the protein complexes to better show complex subunit stoichiometries. However, I suggest the authors use unique colors per different type of subunit in the complex. Now several different subunits are all purple or blue, and it’s still hard to connect it to the bar diagram with RPF data.

Response: As suggested by the reviewer, we revised the Figure 2. The stoichiometry of the multimeric protein complex is shown with unique colors per different types of subunits in the complex.

FIG 2. Translational status of WL pathway under autotrophic growth conditions. (A) RNA expression level and translation level of hydrogen-dependent carbon dioxide reductase (HDCR) under autotrophic growth conditions. Illustration and subunit ratio in HDCR

complex were shown. For CO₂ reduction, electrons are either provided by reduced ferredoxin or by the HydA-HydB3 subunits, where H₂ oxidation occurs. (B) Comparison between the mRNA (RPKM_{RNA}) and translation (RPKM_{RPF}) levels of the methyl-branch of the WL pathway under autotrophic growth conditions. Illustration and subunit ratio in MTHFR complex are shown. (C) RPF level for the carbonyl-branch of the WL pathway under autotrophic growth conditions. Illustration and subunit ratio in ACS/CODH and MET complex are shown. (D) Correspondence between the published protein abundance of *C. ljungdahlii* (27) and the translation ratio of carbonyl-branch proteins. The reported emPAI value for each protein was used for determining the protein content (mol%) using formula: $\text{emPAI} / \sum(\text{emPAI}) \times 100$. *C. ljungdahlii* values are plotted against the relative translation levels of *A. woodii* (Pearson's $r = 0.90$).

Q3. Also, the authors commented on my labelling experiment suggestion ((Q3)) that they did not get it. So, I will try to explain this better. I suggest the authors do grow *A. woodii* and the *E. limosum* (with and without overexpressed GCV) with labelled ¹³CO₂ and ¹²C fructose under heterotrophic conditions. If the methylene and formyl-THF are mostly produced through glycine cleavage these C1-moieties will come from fructose and hence (easy to measure) proteinogenic amino acids which have C1-precursors will be fully unlabelled. If these C1-carbons are coming from the methyl-branch of the Wood-Ljungdahl pathway they will lead to incorporating labelled ¹³CO₂ and hence once labelled methionine and histidine. E.g. Histidine obtains one carbon from formyl-THF (in the biosynthesis of ATP, which is incorporated in histidine, sorry I mentioned wrongly before it come from methylene-THF, but this does not matter, it's the same branch). Another possible amino acid to look at is methionine, which has a carbon coming from methylene-THF. I am not sure if the authors will manage to do such an experiment and it's not essential, but it could be very informative.

Response: Thanks for the good suggestions. We agree with the reviewer's suggestion that the ¹³C labelling experiments would be very informative to understand the metabolic pathway. However, there are several reasons that we did not perform ¹³C labelling experiments. First, the experimental design suggested by the reviewer is different from the heterotrophic growth condition (fructose with N₂ gas condition, 200 kPa) conducted in this study. This looks closer to mixotrophy, defined as the concurrent utilization of organic (sugars) and inorganic (CO₂) substrates in a single organism. Second, under heterotrophic growth conditions, not only methylene-THF but also CO₂ is generated from fructose. Thus, we are unable to identify whether or not methylene-THF derives from fructose with ¹³CO₂. Third, since it is difficult to see the function of GCS more clearly in this heterotrophic growth condition, we indirectly confirmed the function of GCS introduced into *E. limosum* more clearly under the CO₂ growth condition. To validate the functional role of the GCS and glycine reductase, the GCS and glycine reductase pathway were introduced in *E. limosum* ΔmetF strain, which is incapable of fix CO₂ via the conventional WL pathway direction but flux for synthesizing methylene-THF, a precursor for glycine, from CO₂ is feasible (**Supporting Fig. 1a**) (*Proc Natl Acad Sci*, doi:10.1073/pnas.1912289117). As hypothesized, ΔmetF pJIR-GCS-GR strain showed cell growth and acetate production, whereas ΔmetF control did not proliferate (**Supporting Fig. 1b–c**). These results indicated that the GCS and glycine reductase pathway is capable of utilizing CO₂ as a carbon source, using hydrogen as an energy source, in the absence of the carbonyl-branch of WL pathway.

Supporting Figure 1. Phenotypical effect of the GCS and glycine reductase in $\Delta metF$ strain. (a). Schematic representation of the reaction involved in the GCS and glycine reductase mediated autotrophic acetogenesis, including GCS (blue) and glycine reductase pathway (red) in $\Delta metF$ strain. (b–c) Cell growth (b) and acetate production (c) of $\Delta metF$ pJIR-No and $\Delta metF$ pJIR-GCS-GR strains were measured under CO₂-H₂ condition. The values are presented as the means of three different biological replicates \pm standard deviation.

Reviewer #2

Q1. P5, L96: “As expected, a positive correlation was observed...”. Why was this to be expected?

Response: Ribo-Seq only captures those mRNAs that are being actively translated. Each Ribo-Seq read corresponds to one translating ribosome. Therefore, in general, the RPF level is dependent on the mRNA level (*Cell*, doi: 10.1016/j.cell.2016.03.014).

Q2. P8, L158-162: If the data found in the parentheses is in Table S2, I’m not sure these values are needed here. Or just include the RPKM_{mRNA} values for each and reference Table S2 for the coefficient of variation.

Response: To help readers understand intuitively without looking for Table S2, CV (coefficient of variation) and RPKM information were described together in the same way as in the previous paragraph.

Q3. P9, L172: “genes were remained to be insignificant.” Does the author mean “genes were determined to be insignificant?”

Response: As the reviewer pointed out, the sentence was revised as follows:
(Page 9, Line 170): Although the changes in the mRNA and RPF levels of pentose phosphate pathway, glycolytic/gluconeogenesis pathway, and TCA cycle are similar, some of the

acetogenesis related genes were determined to be insignificant from the fructose condition to the H₂+CO₂ condition (Fig. 3A).

Q4. P10, L191: “two large protein complexes.” Which two large protein complexes were upregulated?

Response: As the reviewer pointed out, the sentence was revised as follows:

(Page 10, Line 191): In particular, two large protein complexes (ACS/CODH and MET/CoFeSP; median RPKM_{PRF} = 3833.2) were upregulated up to 12-fold at the translation level compared to the HDCR complex (median RPKM_{PRF} = 311.8) under heterotrophic growth conditions, suggesting unbalanced protein expression among the WL pathway proteins.

Q4. P11, L214: “reaction sequences?” Not sure what is meant by reaction sequences. Please clarify.

Response: To make clear, the sentence was corrected as follows:

(Page 11, Line 213): Since the translation level of carbonyl-branch enzymes was higher than that of enzymes from the methyl-branch during growth on fructose, we looked for folate-dependent single-carbon reaction sequences that could replenish the carbon in the methyl-branch.

Q5. P19, L391: ‘ribosome assemble’ should read ‘ribosome assembly?’

Response: We thank the reviewer for pointing out the shortcoming. As the reviewer pointed out, the sentence was revised as follows:

(Page 18, Line 386): The RPF read density observed in the 5'-UTRs indicates the presence of the attached 70s ribosome and that ribosome assembly may occur upstream of the SD sequence.

Q6. P18, L381-389: The authors mention two GCS-related genes *lpdA1* and *gcvH1* and how they were upregulated under autotrophic growth conditions. The authors describe how these genes are highly conserved in a specific set of acetogens, yet recent work has shown that these two genes are not highly conserved within the *Acetobacterium* genus (Ross et al., 2019-Defining genomic and predicted metabolic features of the *Acetobacterium* genus). Please include findings from this study in light of the findings of Poehlein and coworkers, or remove the sentence pertaining to the conservation of these genes, as it is misleading as written.

Response: We thank the reviewer for pointing out. To prevent misleading about the conservation of *lpdA1* and *gcvH1* genes, the sentences were deleted and revised as follows:

(Page 18, Line 382): Under autotrophic growth conditions, it was also reported CO₂ can be converted into acetyl-CoA and acetyl-phosphate *via* the functional cooperation of the WL

pathway, GCS, and glycine reductase in *C. drakei* and *E. limosum* (28).

Q7. The authors argue that autotrophic growth is energy limited. DSMZ 135 medium contains 2 g/L yeast extract. Does this change the available ‘energy’ in the system? What happens when less YE is utilized?

Response: Previously, yeast extract was shown to modulate the growth and metabolism of several acetogenic species. Previous studies have evidenced the role of some vitaminic factors provided by yeast extract as essential growth factors in some acetogenic species such as *Clostridium* strain and *Eubacterium limosum* (*Appl Environ Microbiol*, doi: 10.1128/aem.61.9.3466-3467.1995; *Curr Microbiol*, doi: 10.1007/s002849900358; *Appl Environ Microbiol*, doi: 10.1128/aem.42.1.12-19.1981). Whether non-vitaminic factors provided by yeast extract are used as energy sources in autotrophic acetogenesis metabolism has not been elucidated so far. In addition, the energy source of yeast extract, whether present or not, does not significantly affect the autotrophic growth of *Acetobacterium woodii*. The free energy change available to the cells is that of the reaction: $4 \text{H}_2 + 2 \text{CO}_2 \rightarrow 1 \text{ acetate} + \text{H}_2\text{O}$. This is -95 kJ/mol under standard conditions and approximately around 25 kJ/mol at environmental (low pH₂) conditions. This reaction allows for the synthesis of only about 0.3 mol of ATP. This is the gain that is independent of the medium composition as long as it does not affect the three variables H₂, CO₂ and acetate. Depending on the media composition, the method of using this ATP can be different in the cells. In complex media containing yeast extract, biosynthesis costs are low, resulting in faster growth. But this does not change the free energy change of the autotrophic acetogenesis reaction.

July 9, 2021

Prof. Byung-Kwan Cho
Korea Advanced Institute of Science and Technology
Department of Biological Sciences
291 Daehak-ro, Yuseong-gu
Daejeon 305-701
Korea (South), Republic of

Re: mSystems00696-21R1 (Genome-scale analysis of *Acetobacterium woodii* identifies translational regulation of acetogenesis)

Dear Prof. Byung-Kwan Cho:

Thank you for submitting your revision. It is good to see that all remaining comments of the reviewers have been carefully addressed.

Your manuscript has been accepted, and I am forwarding it to the ASM Journals Department for publication. For your reference, ASM Journals' address is given below. Before it can be scheduled for publication, your manuscript will be checked by the mSystems senior production editor, Ellie Ghatineh, to make sure that all elements meet the technical requirements for publication. She will contact you if anything needs to be revised before copyediting and production can begin. Otherwise, you will be notified when your proofs are ready to be viewed.

As an open-access publication, mSystems receives no financial support from paid subscriptions and depends on authors' prompt payment of publication fees as soon as their articles are accepted. =

Publication Fees:

- Minimum resolution of 1280 x 720
- .mov or .mp4. video format
- Provide video in the highest quality possible, but do not exceed 1080p

- Provide a still/profile picture that is 640 (w) x 720 (h) max
- Provide the script that was used

We recognize that the video files can become quite large, and so to avoid quality loss ASM suggests sending the video file via <https://www.wetransfer.com/>. When you have a final version of the video and the still ready to share, please send it to Ellie Ghatineh at eghatineh@asmusa.org.

Sincerely,

Marnix Medema
Editor, mSystems

Journals Department
Table S3: Accept
Fig. S4: Accept
Fig. S1: Accept
Table S2: Accept
Fig. S3: Accept
Text S1: Accept
Fig. S5: Accept
Table S1: Accept
Fig. S2: Accept